# Frequency of Use of Added Sugar, Salt, and Fat in Infant Foods up to 10 Months in the Nationwide ELFE Cohort Study: Associated Infant Feeding and Caregiving Practices

**DOI:** 10.3390/nu11040733

**Published:** 2019-03-29

**Authors:** Marie Bournez, Eléa Ksiazek, Marie-Aline Charles, Sandrine Lioret, Marie-Claude Brindisi, Blandine de Lauzon-Guillain, Sophie Nicklaus

**Affiliations:** 1Centre des Sciences du Goût et de l’Alimentation, AgroSup Dijon, CNRS, INRA, Univ. Bourgogne Franche-Comté, 21000 Dijon, France; marie.bournez@inra.fr (M.B.); elea.ksiazek@chu-dijon.fr (E.K.); marie-claude.brindisi@chu-dijon.fr (M.-C.B.); 2Centre Hospitalier Universitaire de Dijon, Hôpital d’Enfants, Pediatrics, 21079 Dijon, France; 3INSERM, UMR1153 Epidemiology and Biostatistics Sorbonne Paris Cité Center (CRESS), Early life research on later health Team (EAROH), F-75014 Paris, France; marie-aline.charles@inserm.fr (M.-A.C.); sandrine.lioret@inserm.fr (S.L.); blandine.delauzon@inserm.fr (B.d.L.-G.); 412 rue de l’Ecole de Médecine, Paris Descartes University, 75006 Paris, France; 5Centre Hospitalier Universitaire de Dijon, Hôpital d’Enfants, Endocrinology, Nutrition, 21079 Dijon, France

**Keywords:** complementary feeding, breastfeeding, ELFE cohort, sugar, salt, fat, infant caregiving practices, infant feeding practices

## Abstract

The consumption of sugar, salt, and fat in infancy may influence later health. The objective of this study was to describe the frequency of use of added sugar, salt, and fat during the complementary feeding period and the associated infant caregiving practices. Data were obtained from a monthly questionnaire filled by parents for 10,907 infants from the French Etude Longitudinale Française depuis l’Enfance (ELFE) cohort. A score of frequency of use (SU) for added sugar, salt, and fat (oil, margarine, butter, and/or cream) was calculated from the age at complementary feeding introduction (CFI) to the 10th month. Associations between the SU of each added ingredient with infant feeding and caregiving practices were studied with multivariable linear regressions adjusted for familial characteristics. Only 28% of the parents followed the recommendation of adding fat and simultaneously not adding sugar or salt. Breastfeeding mothers were more prone to add sugar, salt, and fat than non-breastfeeding mothers. CFI before four months was positively associated with the SU of added sugar and salt and negatively associated with the SU of added fat. The use of commercial baby food was negatively related to the SU of added salt and fat. The use of these added ingredients was mainly related to breastfeeding, age at CFI, and use of commercial food, and it was independent of the household socioeconomic characteristics.

## 1. Introduction

Current European and French guidelines recommend that complementary foods should be introduced between four and six months old [1,2], while the World Health Organization (WHO) recommends their introduction at six months old [3,4]. Age of introduction among other complementary feeding (CF) practices influence infants’ nutritional status and may have short- and long-term effects on health outcomes, especially regarding growth, obesity, and allergies [5,6,7,8,9]. For example, in the ALSPAC (Avon Longitudinal Study of Parents and Children) cohort, Brazionis et al. showed that a less healthy transitional diet in infants from six to 24 months old was associated with higher blood pressure at 7.5 years of age. More specifically, in this cohort, Brion et al. also showed that sodium intake in four-month-old infants was positively associated with systolic blood pressure (BP) at seven years of age [10]. High BP in adults is a major risk factor for cardiovascular diseases and is positively related to high salt intake [11,12]. Furthermore, reducing salt intake decreases arterial BP in children as in adults, as well as in the primary and secondary prevention of hypertension [12,13,14]. Although the long-term effect of salt intake in infancy remains debated, several recent studies reported a high salt intake compared to the recommended intake [15,16,17,18] in infants and children, including infants as young as eight months old [10].

Similarly, several studies linked the consumption of added sugars (i.e., sugars added to food and beverages as part of processing and preparation) [19] or free sugars, especially in liquid forms, to an increased risk of cardiovascular diseases in adults [20]. While the link between added and free sugar consumption in infancy and later cardiovascular diseases remains to be proven, Wang et al. reported that in children included in the NHANES (National Health And Nutrition Examination Survey) 2009–2014 survey, free sugar consumption exceeded the recommended limit of 10% total energy intake, even during the first year of age and that this consumption rose rapidly from infancy to two years of age [21].

In other respects, an adequate fat intake (total amount and quality) in infancy is necessary to ensure optimal health (regarding neurological development, body composition, etc.) in the early and later stages of life [22,23]. Although the NHANES studies consistently reported high saturated fat intake in American infants [21,24], in French and European infants, total fat intake, especially regarding total energy and essential fatty-acid intake, seems to be insufficient [23,25,26,27,28,29]. Moreover, another study showed that the fat content of breast milk was inversely correlated with 3–12-month gains in weight, body mass index (BMI), and adiposity [30]. Other studies suggested that a high-fat diet in infancy could later lead to favorable health conditions [31,32,33].

Altogether, in infancy, adding sugar or salt is not necessary to fulfill nutritional needs, while a low-fat diet could have unfavorable effects. Although the link between some dietary patterns in infancy and later health outcomes remains unclear, previous studies showed that dietary exposure to tastes during infancy (such as sweetness, saltiness, or fattiness) may influence food preferences in childhood [34,35]. Those preferences could remain the same until adulthood [36]. Moreover, as dietary patterns in infancy persist through time and food preferences remain stable until adulthood [16,36,37,38,39,40,41], it can be assumed that reducing an infant’s exposure to salt and sugar may promote a healthier diet later on. The current recommendations to caregivers of the French “Programme National Nutrition Santé” (PNNS) or National Program on Nutrition and Health are to avoid adding sugar and salt during the CF period and to add non-fried fats, such as oil, margarine, cream, or butter, to complementary foods after six months of age [42]. This could appear contradictory to the WHO and American guidelines, which state that caregivers should fight against an excessive amount of fat, as well as sugar or salt [4,43]. However, the European Society for Pediatric Gastroenterology Hepatology and Nutrition (ESPGHAN) recommends ensuring an adequate intake of polyunsaturated fatty acids during the CF period [2], which can be attained by adding some non-fried fat to infant foods, as suggested by the French Pediatric Society [27]. Such practices were recommended to improve the quality of infant diets [27], and they are justified by the observation that fat intake in French infants is insufficient during the CF period [23,25,26,29,44]. Of course, the additions of sugar, salt, or fat may not represent the main part of free sugar, salt, or fat intake of the infant diet, but they are directly under the control of the adults providing food to the infant. Indeed, the parents are only able to modify the macronutrient content of the complementary food they offer to their infant by adding ingredients to them. Until now, the specific additions of sugar, salt, and fat in complementary food by parents was never described as with potential familial associated characteristics. Thus, the objective of the present study was to describe the use of added sugar, salt, and fat during the CF period in the French Etude Longitudinale Française depuis l’Enfance (ELFE) cohort and to investigate the associations between the use of these ingredients and early feeding and caregiving practices, accounting for familial characteristics.

## 2. Methods

### 2.1. Study Population

The present analysis was based on data from the French ELFE study, a nationally representative birth cohort that included 18,258 children born in a random sample of 349 maternity units in France in 2011 [45]. All infants who were born in the 320 maternity units who agreed to participate on the 25 selected recruitment days over the four seasons and met the inclusion criteria were eligible. The inclusion criteria were as follows: children born after 33 weeks of amenorrhea (WA), mothers aged ≥18 years who were not planning to move outside Metropolitan France in the next three years, and mothers able to read French, Arabic, Turkish, or English, the most common languages spoken by women giving birth in France. The ELFE study received approval from the Consultative Committee for the Treatment of Information for Health Research (Comité Consultatif sur le Traitement des Informations pour la Recherche en Santé; CCTIRS), the National Data Protection Authority (Commission Nationale Informatique et Libertés; CNIL), and the National Statistics Council (CNIS). Each mother had to sign consent for herself and her child. If the father was present at inclusion, he also provided consent; if not, he was informed about the right to oppose his child’s participation. Fifty-one percent of the contacted parents agreed to their child’s participation. Data were collected in standardized interviews conducted by trained interviewers and by self-completed questionnaires [45].

Each mother was interviewed face-to-face at the maternity ward for medical information regarding her pregnancy, her newborn, and her general characteristics. Additional information was provided by medical records from the obstetric and pediatric files. At two months postpartum, phone interviews were conducted with the mothers and fathers to complete the previously collected information and assess most of the familial characteristics described in Section 2.3.

### 2.2. Explained Variables: Scores of Use of Added Sugar, Salt, and Fat

Information on the infants’ complementary food consumption was collected monthly from three to 10 months postpartum using an online or paper self-reported questionnaire about the consumption frequency of 26 foods and nine supplementary items (called “ingredients” in this study), selected to describe the infant diet during CF [46]. This questionnaire and the imputation method for missing data applied to the feeding survey (food and “ingredient” items) were previously published [46]. The monthly frequency of the consumption of sugar, salt, oil/margarine, or butter/cream added by the parents was recorded on a five-point scale: never introduced (0), introduced sometimes (1), often introduced (2), always or almost always introduced (3), does not know (4). The latter value (4) was considered as a missing value. Notably, the food or the type of preparation in which the ingredient could be added was not collected in the feeding survey.

The present study focuses on the addition of sugar, salt, and fat by the parents during the CF period, from three to 10 months of age. In this study, the term “added sugar” does not refer to the “added sugars” as defined by the WHO but only to the sugar that the parents could have added in the complementary foods for their infant. The same definition was used for added salt. Added fat use was defined as the oil/margarine or butter/cream added to the complementary foods. Firstly, an infant was considered a “user” of a considered ingredient for a given month when his/her parents declared having introduced this ingredient at least “sometimes”, i.e., the 1, 2, or 3 values on the five-point scale. Then, for each infant, a score of frequency of use (SU) of each added “ingredient” was created by calculating the mean frequency of use. Only months without missing data on the period of interest were used to calculate the corresponding score. A score equal to zero meant that the considered ingredient was never added from the complementary feeding introduction (CFI) age until 10 months. The maximal possible score was 3, indicating that the considered ingredient was always added when asked during the follow-up.

### 2.3. Potential Determinants of the Use of Added Sugar, Salt, and Fat

#### 2.3.1. Informational Sources about Infant Caregiving Used by the Mothers

Mothers were asked to report the type of informational sources about child caregiving they had used in the two first months of life from several sources. For each source, they could answer whether they used it or not. Four information types were then defined as follows: maternal personal experience, health professionals (doctor, nurse, midwife), media (book, press newspaper, television (TV), radio, internet) and family (maternal and paternal grandmothers, child’s father or the partner, family member, friend). For each group, the modalities of response were yes or no.

#### 2.3.2. Infant Caregiving Practices

At two months postpartum, some caregiving practices were assessed. Firstly, the mother’s reaction when the child did not eat much (whether she was breastfeeding or formula-feeding) was evaluated. The exact question was, “what do you do when he (she) sucks little [the breast] or does not finish his (her) bottle, and he (she) is not sick at this moment?” and the possible answers were categorized as follows: “did not insist”, “insisted or tried later”, “never happened”. The second caregiving practice involved maternal concerns about the child’s health, and the answers were categorized as follows: “no concern”, “concerns about feeding issue”, “concerns about another issue than feeding”.

Those practices were selected because they could reflect the quality of the mother–infant relationship concerning feeding. They also may reflect some maternal attitudes and reactions to infant appetite variations or minor troubles (such as nausea or regurgitations) during the exclusive milk feeding period and later during the CF period.

#### 2.3.3. Infant Feeding Practices

Information regarding milk feeding practices was collected prospectively and monthly from two to 10 months of age, as described in another study [47]. The duration of any breastfeeding was categorized as follows: never breastfed; 0–2 months; 2–6 months; and at least 6 months of breastfeeding. From the monthly feeding survey filled by the parents, the age at CF introduction (CFI) was calculated as previously described [46] and categorized as follows: during the third and fourth month of life (CFI <4 months); between four and six months of age, i.e., during the fifth month and the sixth month of life, and strictly after six months of age (CFI >6 months). An SU of commercial baby foods, which was collected from the same questionnaire as the use of added sugar, salt, and fat, was constructed by calculating the mean frequency consumption on the period from the age at CFI to 10 months of age for each infant, only using months without missing data for this period.

#### 2.3.4. Infant and Parental Health and Socioeconomic Characteristics

The newborn characteristics considered in this study were sex, birth rank, twin birth, gestational age, birth weight, and medical diagnosis of an allergy to cow milk. Maternal health-related factors included prepregnancy BMI, smoking status at two months postpartum, and mode of delivery.

The parental socioeconomic characteristics considered in this study were as follows: maternal age at delivery, maternal marital status, maternal education level, parental age difference, paternal education level, family monthly income, and parental country of birth.

### 2.4. Sample Selection

Of the 18,258 initially included infants, 17,856 were eligible (Figure 1). We randomly selected one twin in the case of twin birth (*n* = 277) to avoid family clusters. We then excluded infants with little or no information (*n* = 5654) regarding CF and then those without calculable age at CFI after the imputation of missing data (*n* = 994), leading to a maximal sample of 10,931 infants. The SUs of sugar, salt, and fat were calculable for 10,907 infants. Missing data in variables included in the multivariable models led to a final analyzed sample of 10,159 infants.

### 2.5. Statistical Analysis

To provide representative statistics of all French infants born in 2011, we weighted data related to the use of added sugar, salt, and fat (monthly use and SU) to account for the sample design, nonparticipation in inclusion, and nonresponse to the infant feeding questionnaire. Weighting also included calibration on margins from the state register’s statistical data and the 2010 French National Perinatal study [48] on the following variables: maternal age, region, marital status, migration status, level of education, and parity. This weighting was calculated for the sample that completed the infant feeding questionnaire at least once from three to 10 months.

The associations between the SU of added sugar, salt, and fat, along with information sources, and caregiving and feeding practices were assessed by linear regressions, separately for each ingredient. A study of the residuals of multivariate models was performed to confirm the hypothesis of normality and homogeneity of the variances. Potential confounding factors were selected using the directed acyclic graph method (www.dagitty.net) [49]. Consequently, all models were adjusted for birth rank, maternal characteristics (prepregnancy BMI, age at delivery, education level), paternal characteristics (age difference with mother, education level), parental country of birth, and characteristics related to study design (maternity unit size, season of inclusion, and residential region). The effects associated with the adjustment factors are reported in Appendix A. The significance level was set at *p* < 0.05. All analyses were performed using SAS V9.3 (SAS, Cary, NC, USA).

## 3. Results

### 3.1. Characteristics of the Study Population

The characteristics of the study population with SUs available for all ingredients (*n* = 10,907; included participants) and of the population with SUs not available for at least one ingredient (excluded from the analysis) are described in Table 1. Parents of the included participants were older and more educated, were more often born in France, and had a higher income than those excluded from the analysis. Mothers of the included participants more often had a normal BMI, were less likely to have smoked after birth, and were more likely to have breastfed longer than two months.

### 3.2. Use of Added Sugar, Salt, or Fat

The rate of parents who added each ingredient at least “sometimes” increased progressively between the third and the 10th month, calculated on the weighted data (Figure 2). The pattern of evolution of use with age was quite similar for added salt and added sugar; 9.3% of the parents used added sugar and 8.3% used added salt at six months, while 25.1% and 24.5% used added sugar and salt, respectively, at 10 months. The use of added fat (butter, cream, margarine, and/or oil) was more frequent and increased more rapidly with age, from 18.7% at six months up to 39.8% at eight months and 55.5% at 10 months. The mean SUs of sugar and salt over the time spanning the CFI age until 10 months were 0.18 (SD 0.35) and 0.17 (SD 0.38), respectively; the mean SU for added fat was 0.59 (SD 0.71). Of note, 33.2% of parents were users of added sugar (i.e., SU > 0), 27.7% were users of added salt, and 64% were users of added fat. The three recommendations of adding fat, not adding sugar, and not adding salt simultaneously were followed by 28.2% of the parents. Unweighted data were used to compare the SU of added sugar and salt between the added fat users and the non-added fat users. From their age at CFI until 10 months, 3045 (27.9%) parents were nonusers (SU = 0) of any of the three ingredients, while 1681 (15.4%) were users of all three ingredients. The mean SU of added sugar was lower in the subgroup of non-added fat users than in the subgroup of added-fat users (0.10 (SD 0.27) vs. 0.23 (SD 0.38), *p* < 0.0001). The mean SU of added salt was also lower in the subgroup of non-added fat users than in the subgroup of added fat users (0.04 (SD 0.20) vs. 0.24 (SD 0.44), *p* < 0.0001).

### 3.3. Associations between Infant Feeding and Caregiving Practices and the Use of Added Sugar, Salt, and Fat

#### 3.3.1. Use of Added Sugar

As shown in Table 2, the SU of added sugar was positively related to any breastfeeding (regardless of the duration) compared to no breastfeeding and to a CFI before four months compared to CFI between four and six months, but it was not positively related to a CFI after six months. Reporting family or maternal personal experience as informational sources for infant caregiving was positively related to the SU of added sugar, while reporting the media or health professionals as informational sources was not related to the SU of added sugar. Mothers who reported that “it never happened that their infant did not eat much” were less likely to add sugar compared to those who “did not insist” when their child did not eat much. Neither SU of commercial baby foods nor maternal concerns about the child’s health were related to the SU of added sugar.

#### 3.3.2. Use of Added Salt

The SU of added salt was positively related to any breastfeeding, compared to no breastfeeding, and to a CFI before four months compared to a CFI between four and six months. The SU of added salt was negatively related to the SU of commercial baby foods. Mothers who reported “to insist or propose later” when their infant did not eat much were more likely to add salt compared to those who reported that “they did not insist” when their child did not eat much. Neither informational sources reported by the mother nor maternal concerns about the child’s health were related to the SU of added salt.

#### 3.3.3. Use of Added Fat

The SU of added fat was positively related to any breastfeeding duration, with a linear trend (*p* < 0.001 for trend), and negatively related to a CFI before four months, compared to a CFI between four and six months. The SU of added fat was negatively associated with the SU of commercial baby foods. Using the media as an informational source for infant caregiving was positively related to the SU of added fat, whereas using family as an informational source was negatively related to the SU of added fat.

Maternal concerns about child health, maternal reaction when the child did not eat much, and reporting health professionals or maternal personal experience as informational sources in child caregiving were not related to the SU of added fat.

## 4. Discussion

### 4.1. Main Findings

To our knowledge, our study is the first to assess the use of added sugar, salt, and fat during the CF period and their associated factors in a nationally representative population. The first main result is that approximately one-third of infants received added sugar and salt over their CF period, which is not recommended by public health nutrition guidelines; however, added sugar and salt were not often used. The addition of fat, which is recommended at this age by the French guidelines [1], was only observed in 64% of the infants, but the SU of added fat was higher than those of added sugar and salt, reflecting a more regular use of added fat. Interestingly, 28% of the parents followed the French recommendation of adding fat, but not sugar or salt.

The second main result is the demonstration of the familial characteristics associated with the frequency of use of these ingredients. Any breastfeeding was strongly and positively associated with the use of added fat, sugar, and salt, while a CFI <4 months was positively related to the use of added sugar and salt and negatively related to the use of added fat. The use of ready-made baby foods was negatively related to the use of added fat and salt. Family or personal experiences as informational sources were positively related to the use of added sugar. Using the media or the family as informational sources for caregiving was related to the use of added fat in opposite ways. When the child did not eat much at two months, compared to mothers who “did not insist”, those who “insisted or proposed later” added more salt, while those who reported that “it never happened” added less sugar.

### 4.2. Use of the Ingredients

Concerning sugar and salt, the French PNNS guidelines recommend no addition to infant foods. Here, 24.8% of the parents had never used salt, sugar, or fat. From the beginning of CF until 10 months of life, approximately 35% of the infants had received some added sugar, and approximately 30% had some added salt. This may reflect a lack of diffusion or awareness of the current recommendations.

Concerning fat, the French PNNS advises caregivers to add non-fried fat during CF [1]. This could seem contradictory with guidelines for older children and adults [1,42], which recommend avoiding the addition of sugar, salt, and fat. It could also seem contradictory with the EFSA (European Food Safety Authority) recommendations stating that, after 12 months, the fat contribution to the energy intake should be gradually reduced from 40% to 35–40% of total energy intake up to 36 months [26]. Moreover, the addition of fat in infancy is not explicitly recommended by different public health bodies [2,4,23]. This discrepancy regarding the addition of fat could partly explain why these guidelines are not followed by more parents. In addition, there is no explanation regarding the health benefits of adding fat that is provided with these recommendations, which could have prevented parents from implementing this specific recommendation.

However, the use of added fat is more frequent than the use of added sugar or salt, as expected.

### 4.3. Associations with Feeding and Caregiving Practices

#### 4.3.1. Familial or Cultural Culinary Habits

In this study, the use of added sugar and salt was stronger within the subgroup of added fat users than in the subgroup of non-added fat users. This suggests that adding fat may not exclusively be a result of a willingness to follow the recommendations, but rather a marker of cooking habits, including adding salt, sugar, and fat to complementary foods. Such cooking habits may be inherited through familial or cultural transmission. This is suggested, on the one hand, by the fact that reporting family as an informational source regarding infant caregiving was related to a higher SU of added sugar and fat and, on the other hand, by the fact that the parental country of birth was related to the SU of salt and fat, reflecting cultural culinary practices (Appendix A).

#### 4.3.2. Second or First Babies: Not the Same Practices?

A second- or later-born baby among siblings was also related to a more frequent use of added sugar and salt than in first-born babies and a less frequent use of fat, which could indicate that, in larger families, the family diet is proposed to the infant rather than a dedicated diet, in accordance with previous descriptions [50]. This is also consistent with the findings of Synnott et al., who studied parental perceptions of feeding practices: some parents reported that they followed the guidelines for the first child but “did not follow them as closely for the second one” or that “it is tradition to have children eat from the table early” [51].

#### 4.3.3. Mode of Milk Feeding and Addition of Ingredients

Any breastfeeding was positively related to the use of added sugar and salt. This could appear surprising because longer breastfeeding duration is generally associated with “healthier” dietary patterns [52] and with better compliance with nutritional guidelines [53]. However, our results are also consistent with the results of Yuan et al., who found that breastfeeding duration longer than six months was related to a higher exposure to sweet taste [44] and to a higher carbohydrate intake [28].

However, longer breastfeeding duration was also associated with an increased use of added fat. Therefore, in addition to “healthy eating habits” or the “willingness to follow the recommendations”, this association between longer breastfeeding duration and the use of the three added ingredients to the infant’s diet may reflect different cooking habits in breastfeeding mothers compared to those of non-breastfeeding mothers. Breastfeeding mothers could be more prone to offer homemade foods to their infants, which could explain their more frequent use of added ingredients. Such a dietary pattern described as “longer breastfeeding duration and use of homemade food (and late CFI)” was previously observed in the French EDEN study [52]. Conversely, the use of commercial foods (convenience foods) was shown to be negatively related to age, education, being a woman, having children, and cherishing the naturalness in a Swiss population [54]. Some of these characteristics (older age, higher education level) are related to breastfeeding in the French population [55] and could also determine the choice of homemade food rather than commercial baby food. However, more information about the context and reasons of using commercial baby food versus homemade food appears necessary to better understand if there is a link between their use and the mode of milk feeding. Unfortunately, this level of information was not available in the present study.

One may wonder whether this pattern (longer breastfeeding associated with cooking practices) is specific to French mothers. They were previously described to pay particular attention to the taste discovery for their infant during the CF period [56], contrary to United Kingdom (UK) mothers [57]. This importance of “taste”, explicitly or not, may explain their practice of using added sugar, salt, and fat and may compete with their willingness to follow nutritional recommendations.

#### 4.3.4. Use of Commercial Baby Food or Homemade Food

No relationship between the use of added sugar and the use of commercial baby foods was found, conversely to the observed associations between use of added salt and fat and the use of commercial baby foods. It is difficult to explain this observation in the absence of specific information about the type of food in which the ingredients were added to. Although it may be logical to add ingredients more often to homemade foods than to commercial baby foods, which may be considered to be “adequate” by parents, sugar could be added in some processed foods, such as yogurts. More detailed information about the context of the use of the added ingredients appears necessary to fully understand infant food preparation practices.

We observed that the more the parents used commercial baby foods, the less they added fat to their infant’s diet. This is consistent with previous results [28]. This may indicate that the parents who used commercial baby foods could have believed they were fully nutritionally adequate for the infant’s diet, suggesting they used them without any modification. However, the dietary fat intake of infants in the French sample is certainly too low compared to recommendations [28], even in heavy users of commercial baby foods. For these reasons, it seems that the importance of adding fat to infants’ diet should be further emphasized to parents, even to those who use commercial baby foods [32].

A CFI before four months was positively related to the use of added sugar and salt and negatively related to the use of added fat. This could be explained by the fact that, as the mothers introduced complementary foods at an earlier time, they had more time to also introduce sugar and salt. Another explanation could be that the infants who were introduced to CF before four months may have an “unhealthy dietary pattern”, while those introduced after six months may have a “healthier” diet with a higher compliance to public health nutritional recommendations (which suggest no addition of sugar and salt). This is consistent with the results of Yuan et al., who showed, based on data from another French sample, that a CFI before six months was associated with higher exposure to sweet taste [44].

#### 4.3.5. Recommendations, Media, and Health Professional Advice

The supposed influence of the official public health recommendations cannot be denied, although it cannot be directly assessed here. The mothers who took the media as an informational source were more likely to add fat, while reporting using family as an informational source was associated with a higher SU of added sugar and lower SU of added fat. A 2001 Cochrane review showed that the mass media could be useful in influencing the behavior of health professionals and patients [58]. Surprisingly, whether medical information sources were used did not influence the SU for any of the ingredients, suggesting that health professionals do not deliver consistent information regarding CF practices. This may be due to inconsistent level of knowledge regarding the effect of CF on health outcomes. Clear evidence-based data about the health effect of the consumption of these ingredients at this very specific time window in life may help professionals to emphasize the improvement of fat intake or discourage sugar and salt addition.

#### 4.3.6. Maternal Attitudes toward Infant Feeding

The maternal reaction when the child did not eat much at two months was related to the use of sugar and salt but not fat. The mothers who reported that they “insisted” added more salt than those who “did not insist”, while the mothers who reported that “it never happened that the child did not eat much” added less sugar than those who did not insist. The mothers who reported that “it never happened that the child did not eat much” could be more sensitive to the satiety cues of the infant. Then, even if their infant ate little, they did not consider it as a sign of “eating too little” but as an appropriate regulation of appetite, consciously or not. Those mothers may have paid attention to give to their infant “healthier” products, avoiding added sugar at the CF stage. Alternatively, the infants who “never ate too little” at two months may be “big eaters” during the CF period; thus, the mothers may have tried to limit their energy intake by limiting the addition of sugar. The mothers “who insisted” used more added salt than the mothers “who did not insist”. In the same population, they also introduced CF earlier than those who did not insist [46]. They may have noticed that adding salt increases the intake and acceptance of food in infants and children, as shown in previous studies with children of various ages [59,60].

### 4.4. Strengths and Limitations

#### 4.4.1. Strengths

This study has numerous strengths; this is the first study to assess the use of added sugar, salt, and fat in a nationally-scaled longitudinal study. The relationship between the use of these ingredients and several feeding and caregiving practices—some of which are previously well described (breastfeeding) and some more original (maternal reaction when the child did not eat much, for example)—taking into account the familial characteristics, was possible.

#### 4.4.2. Limitations

Nevertheless, some limitations must be considered to generalize these results. In the ELFE cohort, a significant refusal rate was observed at the inclusion period, because almost 51% of the parents did not consent to participate. This is a common limitation to all longitudinal cohort studies. To address the potential impact of missing data on the age at CFI, an imputation method was applied, which can lead to some biases. However, the imputation method was defined to be as conservative as possible [46], taking into account the longitudinal profile of CFI of the respondents.

Nevertheless, the participants who answered the complementary feeding questionnaire presented selection bias compared with the whole ELFE population. Nonrespondent mothers were younger, less educated, more likely to be born outside France, and more likely to smoke, and they had a higher BMI; that is, they were more likely to represent disadvantaged families, which may lead to underestimations of significance and/or effect sizes. However, to obtain nationally representative data from this specific questionnaire on complementary feeding practices, we weighted the data to take into account the inclusion procedure and biases related to nonconsent and nonresponse. This weighting was applied to calculate population estimates of use of the ingredients but was not used for the multivariable analyses, in order to avoid too many assumptions regarding nonrespondents’ feeding practices. With a sample of almost 11,000 children, our study is powerful enough to draw some reliable conclusions about the frequency of use of the ingredients and the feeding and caregiving practices of parents therein.

The food frequency questionnaire was self-administered; it may, therefore, be prone to a social desirability bias, although no judgmental evaluation was returned to the parents. Many factors appear to have a significant association with the current outcomes, but the effect sizes are often small. Another limitation is that the follow-up stopped at 10 months.

Moreover, no information was available about the amount of sugar, salt, or fat added by the parents, nor about the types of foods to which these ingredients were added.

Nevertheless, this is one of the only reports with detailed information about the use of salt, sugar, and fat during the first year, which corresponds to practices which are directly under the control of the parents and then potentially modifiable through public health intervention, through the formulation of more accurate recommendations in the future.

### 4.5. Practical Implications

More research is needed to describe the context in which salt, sugar, or fat are added, for example, commercial vs. homemade food, and the parental knowledge about the composition of the commercial baby foods.

Moreover, as we observed that the SU of fat was negatively related to the SU of commercial baby food, it could be important to reevaluate the nutritional composition of these products and to verify that they cover the nutritional needs of the infants, especially for lipids.

Finally, more studies are needed about the health outcomes in later life related to lipid intake during the CF period, to ensure the current recommendations are accurate. This could help spread the recommendations to the parents and caregivers, and give them practical ways of reaching the recommended intake.

In conclusion, the French PNNS recommendation of adding fat but not adding sugar or salt to complementary foods is only partially followed. The use of added sugar and salt occurred early in life, but the frequency of their use remained relatively low. The use of added fat also occurred early and was more frequent than the use of added sugar and salt, but approximately 35% of the infants never received added fat during their CF period. Thus, promoting the current recommendation should be enhanced, taking into account the fact that the use of these added ingredients seems to be related to maternal habits or culinary practices. Generic feeding practices (breastfeeding, CFI age) are independent factors of the use of these added ingredients and could be indicators to help healthcare providers or infancy professionals deliver personalized information about the infant’s diet to the families. The ELFE study will offer a great opportunity to assess the impact of these guidelines on health outcomes. This is particularly important in ensuring the accuracy and necessity of the current French recommendations about sugar, salt, and fat during the CF period.

## Figures and Tables

**Figure 1 nutrients-11-00733-f001:**
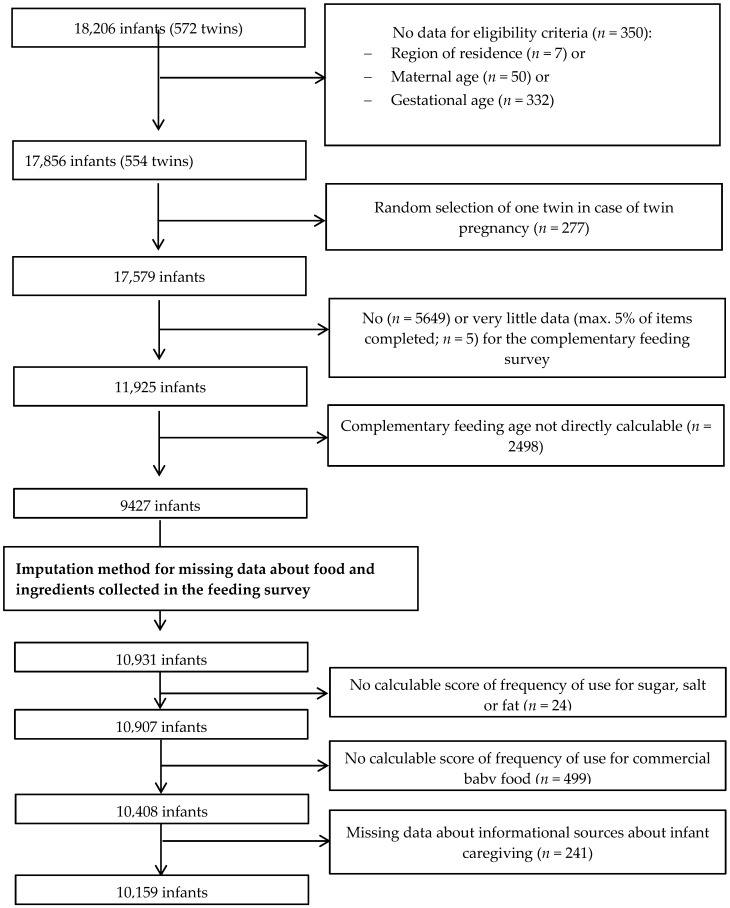
Flowchart of the study population.

**Figure 2 nutrients-11-00733-f002:**
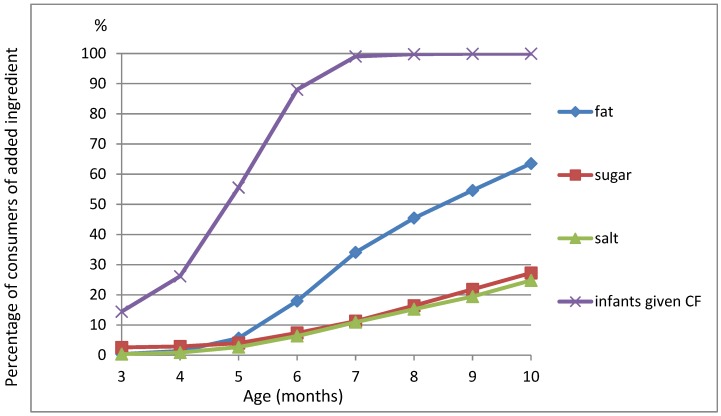
Monthly consumption of added sugar, salt, and fat from three to 10 months old (*n* = 10,907); the percentages are calculated with the weighted data. To be a consumer of one given ingredient is defined as when the use value of the ingredient is at least “sometimes” in the considered month, i.e., for the 1 (sometimes), 2 (often), or 3 (always or almost always) value in the complementary feeding (CF) questionnaire. The upper line (infants given CF) represents the percentage of infants who received complementary foods at the considered age.

**Table 1 nutrients-11-00733-t001:** Characteristics of the study population (with and without data on the three added ingredients).

	*n* (%)			
Characteristics	All	Infants with Calculable Mean Score of Frequency of Use of the Three Ingredients * Included Participants	Infants without Calculable Mean Score of Frequency of the Ingredients: Excluded Infants	*p*-Value **
**All**	17,579	10,907	6672	
**Child characteristics**				
Sex				0.34
Girl	8540 (48.6)	5330 (48.9)	3210 (48.1)	
Boy	9037 (51.4)	5577 (51.1)	3460 (51.9)	
Birth rank				**<0.0001**
1st	7800 (44.4)	5025 (46.1)	2775 (41.7)	
2nd	6278 (35.7)	4042 (37.1)	2336 (33.6)	
≥3	3492 (19.9)	1840 (16.9)	1652 (24.8)	
Twin birth				**<0.0001**
No	17,302 (98.4)	10,771 (98.8)	6531 (97.9)	
Yes	277 (1.6)	136 (1.3)	141 (2.1)	
Gestational age				**<0.0001**
33–37	872 (5.0)	467 (4.3)	405 (6.1)	
37–39	8544 (48.6)	5141 (47.1)	3403 (51.0)	
≥40	8163 (46.4)	5299 (48.6)	2864 (42.9)	
Birth weight (grams)				**<0.0001**
<2500	789 (4.5)	406 (3.7)	383 (5.7)	
2500–3999	15,260 (86.8)	9511 (87.2)	5749 (86.2)	
≥4000	1318 (7.5)	860 (7.9)	458 (6.9)	
Cow’s milk protein allergy reported at 2 months				0.19
Yes	217 (1.2)	141 (1.3)	76 (1.1)	
No	15,257 (86.8)	10547 (96.7)	4710 (70.6)	
**Maternal characteristics**				
Mode of delivery				0.01
Vaginal	14,154 (80.5)	8841 (81.1)	5313 (79.6)	
Caesarean	3137 (17.9)	1882 (17.3)	1250 (18.8)	
Postnatal smoking at 2 months				**<0.0001**
Yes	3129 (17.8)	1764 (16.2)	1365 (20.5)	
No	12,139 (69.1)	8875 (81.4)	3264 (48.9)	
Prepregnancy BMI (kg/m²)				**<0.0001**
<18.5	1359 (7.7)	767 (7)	592 (8.9)	
18.5-24.9	11,191 (63.7)	7258 (66.6)	3933 (59)	
25.0–29.9	3039 (17.3)	1795 (16.5)	1244 (18.7)	
≥30	1730 (9.8)	979 (9)	751 (11.3)	
Age (years)				**<0.0001**
<25	2141 (12.2)	842 (7.7)	1299 (19.5)	
25–29	5479 (31.2)	3376 (31.0)	2103 (31.5)	
30–34	6190 (35.2)	4293 (39.3)	1897 (28.4)	
≥35	3769 (21.4)	2396 (22.0)	1373 (20.8)	
Education level (years)				**<0.0001**
≤9	3567 (20.3)	1530 (14)	2037 (30.5)	
12	2942 (16.7)	1896 (17.4)	1046 (15.7)	
14	3469 (19.7)	2621 (24)	848 (12.7)	
≥15	5776 (32.9)	4655 (42.7)	1121 (16.8)	
**Paternal characteristics**				
Age difference with mother				**<0.0001**
Younger father	3164 (18)	2024 (18.6)	1140 (17.1)	
Father 0–1 years older	4545 (25.9)	3054 (28)	1491 (22.4)	
Father 2–3 years older	3476 (21.3)	2454 (22.5)	1292 (19.4)	
Father 4–7 years older	3684 (21)	2216 (20.3)	1468 (22)	
Father at least 8 years older	1992 (11.3)	1046 (9.6)	946 (14.2)	
Education level (years)				**<0.0001**
≤9	3855 (21.9)	2389 (21.9)	1466 (22)	
12	2738 (15.6)	1995 (18.3)	743 (11.1)	
14	2412 (13.7)	1904 (17.5)	508 (7.6)	
≥15	4420 (25.1)	3632 (33.3)	788 (11.8)	
**Household characteristics**				
Marital status when the child is 2 months				**<0.0001**
Single mother	735 (4.2)	243 (2.2)	492 (7.4)	
Living with someone	15124 (86.03)	10465 (96)	4659 (69.8)	
Monthly income per consumption unit (euros)				**<0.0001**
<600	776 (4.4)	262 (2.4)	514 (7.7)	
600–1099	2713 (15.4)	1406 (12.9)	1307 (19.6)	
1100–1799	6840 (38.9)	4942 (45.3)	1898 (28.5)	
1800–2999	3890 (22.1)	3188 (29.2)	702 (10.5)	
≥3000	773 (4.4)	621 (5.6)	152 (2.3)	
Parental country of birth				**<0.0001**
Mother and father France	13776 (78.4)	9391 (86.1)	4385 (65.7)	
Mother abroad and father France	983 (5.6)	502 (4.6)	481 (7.2)	
Mother France and father abroad	1219 (6.9)	591 (5.4)	628 (9.4)	
Mother and father abroad	1267 (7.2)	327 (3.0)	940 (14.1)	
**Feeding practices**				
Any breastfeeding duration				**<0.0001**
0	4880 (27.8)	2686 (24.6)	2194 (32.9)	
<2 months	4248 (24.2)	2636 (24.2)	1612 (24.2)	
2–6 months	4576 (26)	2765 (25.4)	1811 (27.1)	
≥6 months	3865 (22)	2820 (25.9)	1045 (15.7)	

** After missing data imputation. * Based on chi-square test comparing included infants with excluded infants. Abbreviations: BMI: body mass index. In bold *p* < 0.0001.

**Table 2 nutrients-11-00733-t002:** Multivariate analyses of associations between the score of frequency of use (SU) of added sugar, salt, and fat from the beginning of complementary feeding until 10 months of life, and infant feeding and caregiving practices (*n* = 10,159).

	Sugar		Salt		Fat	
	β (95% CI)	*p*-Value	β (95% CI)	*p*-Value	β (95% CI)	*p*-Value
**Infant feeding practices**						
**Breastfeeding duration**		**0.001**		**0.0004**		**<0.0001**
never breastfed	Reference		Reference		Reference	
0–2 months	0.03 (0.01; 0.05)		0.03 (0.01; 0.05)		0.05 (0.02; 0.09)	
2–6 months	0.03 (0.01; 0.05)		0.02 (0; 0.04)		0.07 (0.03; 0.11)	
≥6 months	0.04 (0.02; 0.06)		0.05 (0.02; 0.07)		0.23 (0.19; 0.27)	
**Age at complementary feeding introduction**		**0.0004**		**0.03**		**<0.0001**
<4 months	0.03 (0.02; 0.05)		0.03 (0.01; 0.04)		−0.09 (−0.13; −0.06)	
4-6 months	Reference		Reference		Reference	
>6 months	−0.02 (−0.04; 0.003)		0.003 (−0.02; 0.03)		0.04 (−0.004; 0.08)	
**SU of commercial baby food**	−0.003 (−0.01; 0.01)	0.46	−0.01 (−0.02; −0.005)	**0.002**	−0.07 (−0.09; −0.06)	**<0.0001**
**Informational sources about infant caregiving**						
**Family ***		**0.02**		0.22		**0.01**
Yes	0.02 (0.003; 0.03)		0.01 (−0.01; 0.03)		−0.04 (−0.06; −0.01)	
**Media ***		0.15		0.18		**<0.0001**
Yes	−0.01 (−0.02; 0.004)		−0.01 (−0.03; 0.01)		0.07 (0.04; 0.10)	
**Health professionals ***		0.60		0.75		0.79
Yes	0.01 (-0.02; 0.03)		−0.004 (−0.03; 0.02)		−0.01 (−0.06; 0.04)	
**Maternal personal experience ***		**0.045**		0.21		0.78
Yes	0.02 (0; 0.04)		−0.01 (−0.03; 0.01)		−0.01 (−0.04; 0.03)	
**Infant caregiving practices**						
**Maternal concern about her child’s health**		0.34		0.23		0.93
No concern	Reference		Reference		Reference	
Feeding issues	0.01 (−0.01; 0.02)		−0.01 (−0.02; 0.01)		−0.01 (−0.04; 0.02)	
Other issues	−0.04 (−0.10; 0.03)		0.01 (−0.06; 0.08)		−0.01 (−0.14; 0.12)	
**Maternal reaction when her child did not eat much**		**0.01**		**0.004**		0.98
“insisted or proposed later”	0.01 (−0.004; 0.03)		0.02 (0.01; 0.04)		0 (−0.03; 0.03)	
“it never happened”	−0.02 (−0.04; −0.004)		−0.01 (−0.03; 0.01)		−0.002 (−0.04; 0.03)	
“did not insist”	Reference		Reference		Reference	

In bold: significant variable (*p* < 0.05). The SU of an ingredient was calculated as the mean frequency of use from the complementary feeding introduction to 10 months. Each month, the parents ranked the use as 0: never, 1: sometimes, 2: often, 3: always or almost always. Consequently, for a given infant, the potential minimal score could be zero, which means that the infant never received the ingredient from the CFI age to 10 months. The potential maximal score could be 3, which means that the infant always received the considered ingredient at every month of the follow-up. * The reference modality is “no”. Linear regressions also adjusted for birth rank, mother’s prepregnancy BMI, maternal age at delivery, maternal education level, paternal age difference with the mother, paternal education level, parental country of birth, maternity unit size, season of inclusion, and residential region. The *r²* values of each model are 0.02, 0.04, and 0.10 for the SU of added sugar, salt, and fat, respectively. Abbreviations: CI: confidence interval; SU: score of frequency of use.

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
