# Peer review of "Frequency of Use of Added Sugar, Salt, and Fat in Infant Foods up to 10 Months in the Nationwide ELFE Cohort Study: Associated Infant Feeding and Caregiving Practices"

_nutrients, 2019, doi:10.3390/nu11040733_

Round 1
Reviewer 1 Report
The manuscript I received to review did not include references as numbered in text.
There is no reference to WHO Infant feeding guidelines (or ESPGHAN)in this paper .
There is no discussion in the introduction of when complementary food introduction should occur.
The scoring by 'score of use' gives no indication to the amounts given only the frequency of use.
A major issue I believe needs resolution is within 2.2 Line 39 "Only 28%of the parents followed the recommendation of adding fat and simultaneously not adding sugar or salt" - No reference and this needs validation. There is no description of what 'fat' is being described. Are you talking about saturated, unsaturated or trans fats? This does not match the French National Nutrition and Health Programs dietary guidelines.
The introduction needs to be rewritten. Line 83 to 85 "adding non fried fats such as oils, cream or butter" is contrary to what is prescribed in other countries feeding guidelines. The paper needs to state the recommended time of complementary food (it is not 3 months of age!).
2. Methods
Only 1/3rd of the ELFE study participants agreed to participate in this study? This needs more discussion and how it effects the participation in this study e.g. socio economic status, literacy.
3.Results
3.1 partially describes the previous point as a result.Selection bias.
3.2 Line 227-228 again need to review recommendation of adding fat. This entire paragraph needs reviewing in context of international guidelines.
4. Discussion
Line 286 ? nationally representative (in view of 2 above?)
Line 291 and 292 review as previously suggested.
Line 295-296 ready made baby foods may include additives of salt, sugar and fat
Use of ingredients some valid points but this needs rewriting in consideration of International infant feeding guidelines.
Also accuracy of parental report needs mentioning.
Line 387 :...suggesting that health professionals do not deliver consistent information regarding CF practices". This point is extremely important but also need to note evidence based practice.
Conclusion
Rewrite in context of evidence based guidelines for infant feeding.
Important subject although this paper needs revising with inclusion of international guidelines.
Author Response
Response to Reviewer 1
The manuscript I received to review did not include references as numbered in text.
They have been included in the revised version. We really apologize for this mistake.
There is no reference to WHO Infant feeding guidelines (or ESPGHAN)in this paper .
Those references were cited but you could not see them as we did not include them in the manuscript you have received. They were initially numbered as references 39, 45 and 46. They are now available earlier in the introduction of this manuscript (cf. following point)
There is no discussion in the introduction of when complementary food introduction should occur.
We added this sentence line 39 to 41: “Current European and French guidelines recommend that complementary food should be introduced between 4 and 6 months old, while the WHO recommend their introduction at 6 months old.”
The scoring by 'score of use' gives no indication to the amounts given only the frequency of use.
We recognize that this word can be a bit confusing. In fact, no amounts of added ingredients were available in the collected data. Then we have changed the expression score of use by “score of frequency of use” and we have replaced “Use of” by “Frequency of use” in the title.
A major issue I believe needs resolution is within 2.2 Line 39 "Only 28%of the parents followed the recommendation of adding fat and simultaneously not adding sugar or salt" - No reference and this needs validation.
This point is one of our results, and is in the abstract, in which we can not add references or validation. But you will find more explanation regarding this observation in the discussion lines 286-290 and 302-306.
There is no description of what 'fat' is being described. Are you talking about saturated, unsaturated or trans fats? This does not match the French National Nutrition and Health Programs dietary guidelines.
In fact, we were not able to distinguish the type of fatty acids or type of fat used. As already described in the methods paragraph, added fat referred here to butter, cream and/or oil added by the parents to their infant’s food. We added this description in the abstract line 25-26.
The French National Nutrition Health Program dietary guidelines state that after 6 months, parents can add non fried fat in infant food [1], as you can also read it in the introduction lines 78-80.
The introduction needs to be rewritten. Line 83 to 85 "adding non fried fats such as oils, cream or butter" is contrary to what is prescribed in other countries feeding guidelines.
You are absolutely right. Some guidelines, as those of the WHO and the American Academy of Pediatrics, state that excessive intake of fat should be avoided, especially by introducing whole cow milk after one year old [2]. The ESPGHAN recommend that an adequate amount of polyinsaturated fatty acids should be insured and that saturated fat should be avoided, but offer no precision how this can be attained [3].
To note, some studies showed that in France, the total intake of fat in the infants diet may be insufficient [4, 5]. Then the French National Nutrition Health Program state that after 6 months, parents can add non fried fat in infant food [1].
All these aspects have been added or discussed more precisely in the text to clarify the finding regarding the different recommendations across the world.
The paper needs to state the recommended time of complementary food (it is not 3 months of age!).
Concerning this point, we added the recommendations line 39 to 41. But, as we showed in our previous paper published in Maternal and Child Nutrition, 2017 Bournez et al., about 25% of the French parents in ELFE introduced complementary feeding before 4 months of age. We are aware that this is not the recommended age of CF. As we wanted to describe the actual practices in infant feeding; we began the collection of data about added ingredients at 3 months of age assuming that some parents may initiate CF before the recommended age. We hope that the clarification of CF age at the beginning of the paper will help clarify this point throughout the article.
2. Methods
Only 1/3rd of the ELFE study participants agreed to participate in this study? This needs more discussion and how it effects the participation in this study e.g. socio economic status, literacy.
Thank you for this comment, but in reality, 51% of the eligible parents agreed to be included in the ELFE study. Then, data about complementary feeding and added ingredients were available for about 51.6% of the 18 258 infants followed in this cohort. Thanks to our imputation method, we could analyze date of 55.6% of the participants of ELFE study, which is quite huge for a longitudinal study, and was large enough to analyze some original factors, even in categories with fewer individuals. This aspect is now discussed lines 430-436.
3.Results
3.1 partially describes the previous point as a result. Selection bias.
You are right, and we have taken this in account and discussed this point in the paragraph 4.2. Limits.
3.2 Line 227-228 again need to review recommendation of adding fat. This entire paragraph needs reviewing in context of international guidelines.
We added precision about what is added fat in the lines 234-235.
We hope that the changes in the introduction may be sufficient to clarify those results about the addition of fat (butter, cream, oil or margarine) in infant food by the parents, given the context of the French practices and guidelines about fat intake in infants.
4. Discussion
Line 286 ? nationally representative (in view of 2 above?)
We cannot deny that there are numerous selection biases in our study. However, data for the ELFE study were actually collected throughout the whole French territory. Then the representativity of the French population is discussed below.
A significant refusal rate was observed at the inclusion period, because almost 51% of the parents did not consent to participate. This is a common limitation to all longitudinal cohort studies. The parents of included children committed themselves and their children for a planned follow-up of 20 years. Such commitment could have hindered their participation, especially because the consent was asked in the 2 to 3 days after birth in the maternity unit.
To address the potential impact of missing data on the age at CFI, an imputation method was applied, which can lead to some biases. However, the imputation method was defined to be as conservative as possible, taking into account the longitudinal profile of CFI of the respondents. We indeed observed that the results of the analyses conducted on our non-imputed data led to conclusions consistent with those drawn from the imputed data set (full results available on request).
Nevertheless, the participants who answered the CFg questionnaire presented selection bias compared with the whole ELFE population.
Nonrespondent mothers were younger, less educated, more likely to be born outside France, and more likely to smoke, and they had a higher BMI; that is, they were more likely to represent disadvantaged families, which may lead to underestimations of significance and/or effect sizes. However, to obtain nationally representative from this specific questionnaire on CFg practices, we weighted the data to take into account the inclusion procedure and biases related to nonconsent and nonresponse. This weighting was applied to calculate population estimates of use of the ingredients but was not used for the multivariable analyses, in order to avoid too many assumptions regarding nonrespondents' feeding practices. With a sample of almost 11,000 children, the categories depicting the disadvantaged families were still large enough to calculate significant estimations of their relationship with the age at CFI. Therefore, our study is powerful enough to draw some reliable conclusions about those categories and the feeding practices of parents therein.
We change the discussion paragraph line 430 to line 448 to address these points. We hope that these changes make our manuscript more accurate.
Line 291 and 292 review as previously suggested.
This comments probably related to the mention of parents who added fat as recommended. See response above, and modifications in the introduction.
Line 295-296 ready made baby foods may include additives of salt, sugar and fat
Of course, you are totally right, but unfortunately it was not possible during this cohort study to collect detailed information such as amount or commercial baby food composition. We only collected the monthly frequency of use of these products. This is why we chose to study a parental feeding practice: adding some ingredients to infant food (whether homemade or commercial), rather the total amount of sugar, salt or fat given to these infants, since this information was not available. We also have considered that those practices are directly under the control of the parents and that better understanding those practices could help to formulate more accurate recommendations, based on actual practices, to improve infants diet. We have clarified this in the discussion and in the practical implication section of the discussion (lines 455-458 and 460-465).
Use of ingredients some valid points but this needs rewriting in consideration of International infant feeding guidelines.
We hope that the changes we have made will convince you about the relevance of our results and discussion;
Also accuracy of parental report needs mentioning.
We are aware of this potential limitation of our study and mention it line 449-450
Line 387 :...suggesting that health professionals do not deliver consistent information regarding CF practices". This point is extremely important but also need to note evidence based practice.
Thank you very much for this comment. We were surprised about this point, and are convinced that more evidence based results may help health professionals to spread the recommendations, although studies about it may lack. We added some information line 466 to line 469. This point has not been much studied in France so we emphasize the lack of evidence based practice.
Conclusion
Rewrite in context of evidence based guidelines for infant feeding.
We believe that our results are relevant regarding the French guidelines and the insufficient intake of total lipids in French infants, and also are important regarding the potential health outcomes related to this insufficiency in lipid intake. But we recognize that the guidelines are different between countries, thus we have specified that we considered French recommendations, lines 471 and 482.
Important subject although this paper needs revising with inclusion of international guidelines.
Thank you for your interest in this topic.
1. Health comes with eating-Group "Food Guides of the French National Program for Nutrition and Health" [http://www.inpes.sante.fr/CFESBases/catalogue/pdf/890.pdf] Accessed
2. Pediatrics AAo: Infant food and feeding. 2018.
3. Fewtrell M, Bronsky J, Campoy C, Domellöf M, Embleton N, Fidler Mis N, Hojsak I, Hulst JM, Indrio F, Lapillonne A et al: Complementary Feeding: A Position Paper by the European Society for Paediatric Gastroenterology, Hepatology, and Nutrition (ESPGHAN) Committee on Nutrition. Journal of Pediatric Gastroenterology and Nutrition. 2017, 64(1):119-132.
4. Bocquet A, Vidailhet M: Nutri-Bebe 2013 Study Part 2. How do French mothers feed their young children? Archives de pediatrie : organe officiel de la Societe francaise de pediatrie. 2015, 22(10 Suppl 1):10s17-10s19.
5. Briend A, Legrand P, Bocquet A, Girardet JP, Bresson JL, Chouraqui JP, Darmaun D, Dupont C, Frelut ML, Goulet O et al: Lipid intake in children under 3 years of age in France. A position paper by the Committee on Nutrition of the French Society of Paediatrics. Archives de pediatrie : organe officiel de la Societe francaise de pediatrie. 2014, 21(4):424-438.

Reviewer 2 Report
This article titled "Use of added sugar, salt and fat up to 10 months in the nationwide ELFE cohort study: associated infant feeding and caregiving practices" is very intriguing and also discusses a very pertinent topic elaborating the characteristics of infant feeding and caregiving practices. It is well designed and well conducted. Also the article is well written.
Author Response
Comments and Suggestions for Authors
This article titled "Use of added sugar, salt and fat up to 10 months in the nationwide ELFE cohort study: associated infant feeding and caregiving practices" is very intriguing and also discusses a very pertinent topic elaborating the characteristics of infant feeding and caregiving practices. It is well designed and well conducted. Also the article is well written.
Thank you for your review. We hope that you will also appreciate the revised version of our manuscript.

Reviewer 3 Report
This is an interesting manuscript with potential key implications to different stakeholders. The paper describes the use of added sugar, salt and fat in homemade food by a long cohort of French parents and its association with infant feeding and caregiving practices. As such, I believe that it provides a significant contribution for researchers, policy makers and health care professionals such as pediatricians, midwifes and nutritionist among other stakeholders. The abstract, the introduction and the methods sections are very well-written and provide a good basis as to what the authors are addressing with their manuscript. Nevertheless, the manuscript suffers from a few limitations. In what follows, I will elaborate on them trying to provide solutions.
My main concern with this paper is that the discussion section is difficult to read, and the main findings are not clear or evident to the reader. Accordingly, I suggest the authors to split this section into two subsections, namely:
1. Practical implications: Here authors need to better describe the main findings of their study and incorporate practical examples of the implications for policymakers, health care professionals and so on.
2. Limitations of the study (Lines 408-419)
Other issues
List of references is not included in the manuscript.
Tittle: I think this should be rephrased to make clear that the use of added sugar, salt and fat is in homemade baby food. Otherwise, readers might be confused by the title.
Author Response
Response to Reviewer 3
This is an interesting manuscript with potential key implications to different stakeholders. The paper describes the use of added sugar, salt and fat in homemade food by a long cohort of French parents and its association with infant feeding and caregiving practices. As such, I believe that it provides a significant contribution for researchers, policy makers and health care professionals such as pediatricians, midwifes and nutritionist among other stakeholders. The abstract, the introduction and the methods sections are very well-written and provide a good basis as to what the authors are addressing with their manuscript. Nevertheless, the manuscript suffers from a few limitations. In what follows, I will elaborate on them trying to provide solutions.
Thank you for your positive comments and your careful reading of the manuscript.
My main concern with this paper is that the discussion section is difficult to read, and the main findings are not clear or evident to the reader. Accordingly, I suggest the authors to split this section into two subsections, namely:
1. Practical implications: Here authors need to better describe the main findings of their study and incorporate practical examples of the implications for policymakers, health care professionals and so on.
2. Limitations of the study (Lines 408-419)
Thank you for your suggestion. We are aware that more research and data are needed to draw some implications, but we did our best to highlight the implication of this work. Furthermore, we made some substantial changes in the discussion, and we hope that it clarified it.
Other issues
List of references is not included in the manuscript.
We apologize for this omission, they have been added.
Tittle: I think this should be rephrased to make clear that the use of added sugar, salt and fat is in homemade baby food. Otherwise, readers might be confused by the title.
Thank you for your comment. As you can see, we changed the title to “Frequency of use of added sugar, salt and fat in infant food up to 10 months” as sugar, salt and fat could be added by the parents as well in homemade food as in commercial baby food.

Round 2
Reviewer 1 Report
Corrections made in response to original review report have addressed all issues identified.